# Layered seawater intrusion and melt under grounded ice

Alexander A. Robel[1], Earle Wilson[2], and Helene Seroussi[3,4]

[1]School of Earth and Atmospheric Sciences, Georgia Institute of Technology, Atlanta, GA, USA
[2]Division of Geological and Planetary Sciences, California Institute of Technology, Pasadena, CA, USA
[3]Jet Propulsion Laboratory, California Institute of Technology, Pasadena, CA, USA
[4]Thayer School of Engineering, Dartmouth College, Hanover, NH, USA

**Correspondence:** Alexander A. Robel (robel@eas.gatech.edu)

**Abstract.** Increasing melt of ice sheets at their floating or vertical interfaces with the ocean is a major driver of marine ice sheet retreat and sea level rise. However, the extent to which warm, salty seawater may drive melting under the grounded portions of ice sheets is still not well understood. Previous work has explored the possibility that dense seawater intrudes beneath relatively light subglacial freshwater discharge, similar to the "salt wedge" observed in many estuarine systems. In this study, we develop a generalized theory of layered seawater intrusion under grounded ice, including where subglacial hydrology occurs as a macroporous water sheet over impermeable beds or as microporous Darcy flow through permeable till. Using predictions from this theory, we show that seawater intrusion over flat or reverse-sloping impermeable beds may feasibly occur up to tens of kilometers upstream of a glacier terminus or grounding line. On the other hand, seawater is unlikely to intrude more than tens of meters through permeable till. Simulations using the Ice-Sheet and Sea-Level System Model (ISSM) show that even just a few hundred meters of basal melt caused by seawater intrusion upstream of marine ice sheet grounding lines can cause projections of marine ice sheet volume loss to be 10-50% higher. Kilometers of intrusion-induced basal melt can cause projected ice sheet volume loss to more than double. These results suggest that further observational, experimental and numerical investigations are needed to determine the conditions under which seawater intrusion occurs and whether it will indeed drive rapid marine ice sheet retreat and sea level rise in the future.

## 1   Introduction

Where ice sheets come into contact with seawater, ice may be lost through melting and dissolution. Increasing ocean-induced ice sheet melt has led to glacier retreat and upstream glacier thinning in Greenland and Antarctica, contributing significantly to global mean sea level rise (Straneo and Heimbach, 2013; Shepherd et al., 2018). The grounding line (the location where glacier ice loses contact with the bed) has traditionally been considered a hydraulic barrier to the intrusion of seawater beneath grounded ice, due to the horizontal hydropotential gradient imposed by the increasing weight of a glacier that thickens upstream. Previous studies have hypothesized that seawater intrusion may only be possible when transient tidal variations in the hydrostatic pressure of seawater at the grounding line either overcome this hydraulic barrier (Walker et al., 2013) or cause it to migrate upstream from the grounding line (Sayag and Worster, 2013). In such models, the compositional difference of seawater and subglacial discharge is not considered, leaving the horizontal variation in overburden pressure as the primary

control on the subglacial intrusion of seawater. However, the interface between freshwater discharge and seawater can have vertical structure where dense seawater intrudes horizontally under the lighter freshwater forming a "salt wedge", as has been observed in estuaries (Geyer and Ralston, 2011), enclosed wastewater outfalls (Adams et al., 1994), and coastal karst channels (Dermissis, 1993).

Wilson et al. (2020) first showed in theory and experiments that layered seawater intrusion is possible in laterally-confined subglacial channels (i.e. Rothlisberger or Nye channels), and may extend several kilometers upstream of glacier termini under realistic conditions. A diverse range of observations have also indicated the possibility for seawater intrusion upstream of grounding lines, including: the absence of a transition in bed reflectivity across the grounding line at Whillans and Kamb ice streams (MacGregor et al., 2011), a large subglacial channel crossing the grounding line of Whillans ice stream imaged with active-source seismic methods (Horgan et al., 2013), a similarly large radar-imaged channel crossing the grounding line at Roi Baudouin Ice Shelf (Drews et al., 2017), and elevated subglacial melt upstream of the grounding line measured by satellite-based interferometric synthetic aperture radar at Thwaites Glacier (Milillo et al., 2019). Such observations provide, at best, plausible arguments for the presence of seawater upstream of the grounding line, but provide few constraints on the composition, temperature, and vertical structure of these potential intrusions.

Understanding the extent of possible seawater intrusion is also directly relevant to projections of mass loss of marine ice sheets, which are highly sensitive to the intensity of ocean melt occurring near the grounding line (Arthern and Williams, 2017; Reese et al., 2018; Goldberg et al., 2019) and the manner in which ocean melt is applied at the grounding line in models (Seroussi and Morlighem, 2018). Ice sheet models which unintentionally include ocean melt kilometers upstream of the grounding line (due to numerical inaccuracies) simulate up to two times more Antarctic ice sheet mass loss in response to climate change over next century (Seroussi et al., 2019). Thus, investigating the possibility that seawater intrusion may drive melt under grounded ice is of first-order importance to the problem of accurately simulating the response of marine ice sheets to climate change.

In this study, we generalize the channelized intrusion theory of Wilson et al. (2020) to a broader range of subglacial hydrology types which encompass most glacier grounding lines and termini (section 2). This generalized theory predicts that for the wide range of parameters describing subglacial hydrology, seawater intrusion can either be suppressed entirely, or extend 10's of kilometers inland (section 3). In a state-of-the-art ice sheet model, we show that the transient rate of ice sheet mass loss depends sensitively on the extent of warm seawater intrusion upstream of marine ice sheet grounding lines (section 4). Finally, we suggest observations which may be useful to better constrain seawater intrusion and subglacial hydrology near the ice-ocean interface, and discuss the implications for simulations of future ice sheet mass loss in response to ocean warming (sections 5 and 6).

## 2   Theory of seawater intrusion distance beneath grounded ice

The goal of this study is to consider the possibility that seawater will intrude under grounded portions of ice sheets. In this section, we consider a generalized theory for the horizontal distance over which this intrusion may occur in the cases of hard (i.e.,

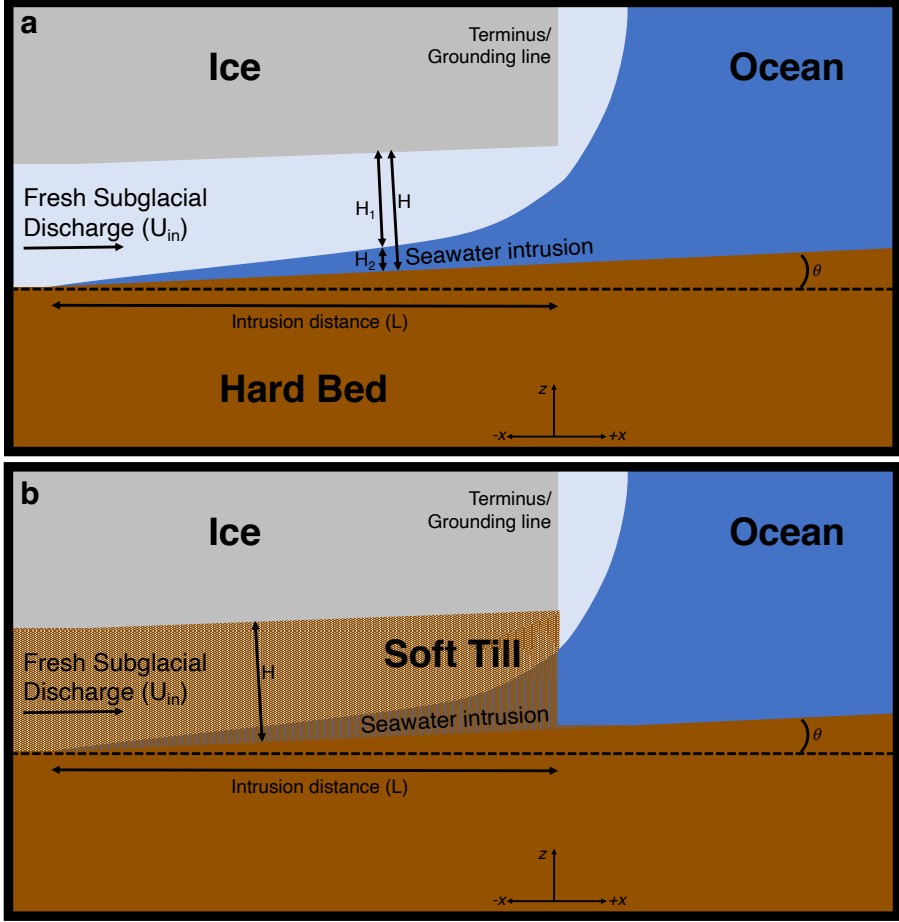

**Figure 1.** Illustration of seawater intruding under grounded ice with relevant variables labeled. (a) Hard bed case. (b) Confined soft bed case. Relative vertical scale exaggerated for clarity.

impermeable) and soft (i.e., permeable beds). In both cases, the intrusion distance depends sensitively on the characteristics of the subglacial hydrological system. In section 3, we discuss the range of predictions from this theory and the relationship to observations of subglacial hydrology.

## 2.1 Hard beds

On hard beds that are impermeable to vertical drainage of water, observations and theory indicate that subglacial hydrology organizes into systems that are either macroporous sheets (i.e., "inefficient" or linked-cavity drainage; Creyts and Schoof, 2009), or channelized (i.e., "efficient" drainage; Rothlisberger, 1972). If such a system is sufficiently permeable in horizontal directions, the flow can be described by the shallow water equations. If the system is less permeable in the horizontal (i.e.

microporous), inertial terms become unimportant and the flow of water is better described by Darcy's law, as we discuss further in section 2.2.

Wilson et al. (2020) originally considered the dynamics of a two-layer shallow water flow for the case of a laterally and vertically confined subglacial channel on an impermeable bed, with free-flowing, fresh subglacial discharge entering a large saltwater body at rest. However, where efficient channelization does not occur, subglacial water flow likely occurs through macroporous sheets (i.e. not laterally confined) which are kept open by water pressure and the influence of large clasts protruding from the bed (Creyts and Schoof, 2009; Hewitt, 2011). We generalize the seawater intrusion theory of Wilson et al. (2020) by considering two-layer macroporous water flow through a vertically confined subglacial sheet over an impermeable bed (illustrated in Fig. 1a). We will show later that the channelized system of Wilson et al. (2020) is a special intermediate case of the generalized theory derived in this study.

We consider a vertically confined, two-layer (fresh and saline) shallow water system, where the grounding line or terminus is located at $X = 0$ and the layers are confined under ice where $X < 0$ (Fig. 1). Mass conservation in both layers is given by

$$\frac{\partial H_1}{\partial t} + \frac{\partial Q_1}{\partial X} = 0 \tag{1}$$

$$\frac{\partial H_2}{\partial t} + \frac{\partial Q_2}{\partial X} = 0 \tag{2}$$

where $H_1$ is the freshwater layer thickness, $H_2$ is the seawater layer thickness, $H = H_1 + H_2$ is the thickness of the full water sheet (i.e. the confined thickness between the bed and ice), and $Q_1 = H_1 U_1$ and $Q_2 = H_2 U_2$ are the area fluxes of water of each layer, with fluid velocities $U_1$ and $U_2$.

In the two-layer macroporous system, flow may be influenced by the relative buoyancy of the two layers, external barotropic pressure gradients, drag from overlying ice and underlying bed, drag at the interface of the two layers, and drag from the embedded obstacles (i.e., clasts). The momentum balance for both layers is then

$$\frac{\partial U_1}{\partial t} + U_1 \frac{\partial U_1}{\partial X} + \frac{1}{\rho} \frac{\partial P}{\partial X} + C_i \frac{|U_1 - U_2|(U_1 - U_2)}{H_1} + U_1^2 \left( C_{ice} \frac{1}{H_1} + C_{obs} \frac{2\phi_1}{\pi d_1 (1 - \phi_1)} \right) = 0 \tag{3}$$

$$\frac{\partial U_2}{\partial t} + U_2 \frac{\partial U_2}{\partial X} + \frac{1}{\rho} \frac{\partial P}{\partial X} - C_i \frac{|U_1 - U_2|(U_1 - U_2)}{H_2} + g' \left( \frac{\partial H_2}{\partial x} + \tan\theta \right) + U_2^2 \left( C_{bed} \frac{1}{H_2} + C_{obs} \frac{2\phi_2}{\pi d_2 (1 - \phi_2)} \right) = 0 \tag{4}$$

where $X$ is the distance from the grounding line (with $X < 0$ upstream of the grounding line), $t$ is time, $\frac{\partial P}{\partial X}$ is the along-flow barotropic pressure gradient, $C_{ice}$, $C_{bed}$, $C_{obs}$ and $C_i$ are the drag coefficients for flow past the ice, bed, obstacles, and layer interface, respectively, $\phi_{1,2}$ are the bulk porosities of the sheet in each layer (defined as the fraction of the water sheet volume occupied by obstacles), $d_{1,2}$ are the mean obstacle diameters in the layers, $g' = g \Delta\rho/\rho_0$ is the reduced gravity for density difference $\Delta\rho$ between the two layers, and $\tan\theta$ is the bed slope. The main difference between this two-layer flow and that described in Wilson et al. (2020) is that the final term in equations 3 and 4 is given by the bulk drag due to flow through a field of obstacles (Rominger and Nepf, 2011), instead of drag due to friction at side walls. $\frac{2\phi_i}{\pi d_i (1 - \phi_i)}$ is related to a length scale of the rough surfaces of the obstacles that are encountered by water flow. We assume hereafter that the drag coefficient is the same for the ice, bed and obstacles ($C_d = C_{bed} = C_{obs} = C_{ice}$) within the macroporous water sheet. We also assume that the obstacles' diameters and porosities are the same in both layers of the water sheet ($d = d_1 = d_2$, $\phi = \phi_1 = \phi_2$). Though these are

simplifying assumptions, the uncertainty on drag coefficients and obstacle properties is sufficiently large that it encompasses any variations that exist within the macroporous water sheet.

As can be seen in equations 3 and 4, the externally-imposed barotropic pressure gradient ($\frac{\partial P}{\partial X}$), which is known as the subglacial hydropotential gradient among glaciologists, acts equally on both layers. A horizontal variation in this barotropic pressure may result from ice surface slope or a difference in slope between the ice base and bed (which produces a gradient in the total subglacial layer thickness, $H$). However, because the fresh and saline layers are vertically layered, barotropic pressure gradients act equally on both layers. Consequently, when the momentum equations are ultimately combined (below), the

pressure gradient falls out of the solution for intrusion distance. This model of subglacial seawater intrusion is focused on the horizontal extent of the saline layer, to which the hydropotential gradient is not directly relevant, and is fundamentally different than previous efforts which focus on the horizontal interface between subglacial discharge and seawater without considering their compositional differences as we do here (through the inclusion of buoyancy and consideration of the potential for vertical layering). Nonetheless, subglacial barotropic pressure far upstream of the grounding line does ultimately set the inflowing

freshwater flow velocity and thus does have an indirect influence on the water velocity scale (as we explore further below).

  The velocity scale is set by the local dimensionless densimetric Froude number, $Fr = \frac{U_1}{\sqrt{g'H}}$, which changes with the local freshwater velocity. Where freshwater discharge is sufficiently slow, the flow of water may be considered subcritical (i.e., $Fr < 1$), and mixing between the two layers is negligible (which would otherwise enter as additional terms in the above equations) as in slow estuarine and subglacial flows, as shown in experiments (Wilson et al., 2020) and observations (Ralston

et al., 2010).

  We proceed by considering the steady state solution to equations 1-4, where the saline layer is at rest ($U_2 = 0$). We also non-dimensionalize the equations 1-4 by making the following substitutions: $h = H_1/H$, $x = C_0 X/H$, $\widetilde{C}_i = C_i/C_0$, $\widetilde{C}_d = C_d/C_0$, $\gamma = 2\phi H/\pi d(1-\phi)$, and $\Theta = \tan\theta/C_0$. $C_0$ is a characteristic scale for drag coefficients and $\gamma$ is a dimensionless parameter capturing the resistance to flow by the macroporous matrix. Combining equations 1-4 to eliminate the barotropic pressure

gradient, and using the relationship $H_2 = H - H_1$, we arrive at the following differential equation for the non-dimensional freshwater layer thickness ($h$)

$$\left(Fr^2 - 1\right)\frac{\partial h}{\partial x} = Fr^2 \left[\widetilde{C}_i(1-h)^{-1} + \widetilde{C}_d(1+\gamma h)\right] - \Theta \tag{5}$$

The Froude number can be written in terms of a Froude number scale

$$Fr = Fr_0 h^{-3/2} \tag{6}$$

where $Fr_0 = \frac{U_{in}}{\sqrt{g'H}}$ is the Froude number of the freshwater inflow upstream where it occupies the entire water sheet ($\lim_{x \to -\infty} U_1(x) = U_{in}$). In the above equation, $Fr$ changes as the thickness, and also velocity of the freshwater layer both change. We note that when $\Theta = 0$, it must be the case that the sign of $\frac{\partial h}{\partial x}$ is set by $Fr^2 - 1$. Since we assume in this study that $Fr \leq 1$, $h$ always increases upstream of the grounding line, requiring that $h = 1$ as $x \to -\infty$.

At the grounding line or terminus ($x = 0$), the freshwater flow becomes unconfined and hence supercritical (i.e. $Fr = 1$ in equation 6), so we set the boundary condition

$$h(x = 0) = Fr_0^{2/3}. \tag{7}$$

The dimensionless intrusion distance, $\ell$, is where the freshwater layer first occupies the entire subglacial water layer thickness ($h(x = \ell) = 1$) upstream from the grounding line, which is also where the saline layer no longer occupies any of the subglacial water layer. In appendix A, we describe how equation 5 can be numerically integrated from $x = 0$, until $h(x = \ell) = 1$, in order to find the intrusion distance, $\ell$. We can also derive analytical approximations for $\ell$ in two limiting cases to describe how the intrusion distance depends on certain parameters describing the subglacial hydrology.

We make three further simplifying assumptions to derive analytical approximations for the intrusion distance. First, we only consider flat beds ($\Theta = 0$), though we will numerically explore how sloped beds modify the intrusion distance in section 3.3. Second, we assume that interfacial drag is negligible ($C_i = 0$), as most estimates of interfacial drag between saline and fresh water layers are $O(10^{-4})$ (MacCready and Geyer, 2010; Geyer and Ralston, 2011), whereas even limiting cases of drag on relatively smooth ice walls are of order $10^{-3}$, and drag is greater yet for realistic subglacial surfaces. Third, we can assume that subglacial freshwater flow is highly subcritical ($Fr_0 << 1$), which is appropriate for places like Antarctica, where there is little injection of surface melt water to the bed and subglacial water flow velocities are low (cm/s or less). With these three assumptions, equation 5 reduces to

$$\frac{\partial h}{\partial x} = \widetilde{C}_d \left(1 - \frac{h^3}{Fr_0^2}\right)^{-1} (1 + \gamma h) \tag{8}$$

where we have re-written the dimensionless parameter groups involving the local Froude number in terms of $h$ and $Fr_0$. This differential equation can be analytically integrated for a solution, but the resulting expression is unwieldy and difficult to interpret, so we consider two end-member cases which provide reasonable lower and upper bounds on expected seawater intrusion distance.

The subglacial clasts which obstruct flow in the water sheet may be spaced far apart and still provide sufficient support to maintain a separation between the ice and bed (or water pressure may maintain this separation on its own, as in Hewitt (2011)). In this scenario, the porosity of subglacial obstacles ($\phi$) is low and so $\gamma h << 1$. Equation 8 can then be integrated exactly to yield an approximation for the intrusion distance for an unobstructed water sheet

$$\ell_u = \frac{1}{4 C_d Fr_0^2}. \tag{9}$$

In this end-member, the dominant drag on the intrusion of seawater into the subglacial water system is the drag of the water against the bed and against the ice. This is also the limiting case of intrusion where the width of channels is much greater than their height, considered previously in numerical calculations by Wilson et al. (2020).

The contrasting limiting case is when clasts which obstruct flow within the water sheet are closely-spaced and so the macro-porosity of the water sheet is high (though not sufficiently so to produce a non-inertial Darcy flow regime, which we consider separately in section 2.2). In such a case, we expect that $\gamma h >> 1$, where equation 8 is integrated to yield an approximation for

the intrusion distance for a macroporous water sheet

$$\ell_p = \frac{1}{3\gamma C_d Fr_0^2}. \tag{10}$$

Re-dimensionalizing $\ell_u$ and $\ell_p$ yields upper and lower bounds on $L_h$ ($L_h = H\ell/C_d$), the dimensional seawater intrusion distance on hard beds, which can be written

 $$\frac{1}{3\gamma}\tilde{L} < L_h < \frac{1}{4}\tilde{L}. \tag{11}$$

We define an "intrusion length scale" for hard beds

$$\tilde{L} = \frac{H^2 g'}{C_d^2 U_{in}^2} \tag{12}$$

which may also be derived from equation 8 using dimensional analysis. As discussed later in section 3, we generally expect $\gamma \lesssim 2$, meaning that the intrusion distance is likely to be a fraction of the intrusion length scale between $1/4$ and $1/6$.

We briefly note that for the case of water flow through a subglacial channel (described in Wilson et al., 2020), the shallow water equations have identical structure, except that $\gamma = 2H/W$, capturing the effect of drag from channel side walls (not obstacle drag as in this model). Observations indicate that the geometry of subglacial channels spans a range from circular (Rothlisberger, 1972), corresponding to $\gamma = 2$ in equation 10, to flat (Nye, 1976) which would correspond to an unobstructed water sheet (equation 9). Thus, although we have focused our discussion of the macroporous water sheet, the two limiting cases of $\ell_u$ and $\ell_p$ also encompass intrusion distances expected in subglacial channels. The approach of calculating two end-member solutions for intrusion distance then provides a general theory that we expect to apply in a wide range of settings with hard beds, regardless of the organization of the subglacial drainage.

In Fig. 2, we evaluate the validity of the assumptions that were made on the way to deriving these analytical approximations for the intrusion distance (purple circles) by comparing them to numerically calculated solutions to equation 5 (black lines). For large $\gamma$ (heavily obstructed seawater intrusion), Fig. 2a shows that $\ell_p$ (Equation 10) is an excellent approximation to the intrusion distance if $Fr_0$ is small. Figure 2b shows that the $\ell_p$ approximation breaks down as $Fr_0$ becomes $O(1)$. At intermediate $\gamma$ (Fig. 2c), $\ell_p$ is off by $O(1)$ as the $\gamma >> 1$ approximation becomes less appropriate. Finally, at small $\gamma$ (unobstructed sheet-like flow), Fig. 2d shows that $\ell_u$ (Equation 9) is a good approximation if $C_i << C_d$. We note that, in general, seawater intrusion distance increases as $\gamma$ and $Fr_0$ decrease (these dependencies are discussed in much more detail in section 3), as predicted by the analytical approximations. Ultimately, where these approximations break down, they are typically too large by factors of $O(1)$. As we discuss later, a lack of constraints on physical parameters which enter into these analytical approximations produce uncertainties in intrusion distance range over 1-4 orders of magnitude. Thus, these approximations are good enough to provide insights into the physical controls on intrusion distance. However, where accuracy is required, numerically solving equation 5 is preferable (as we do in section 3).

## 2.2  Soft beds

Darcy's law is a non-inertial, creeping, incompressible simplification of the Navier-Stokes equations which describes steady-state fluid flow in microporous media. Subglacial water flow can be described solely by Darcy's law where there is a microp-

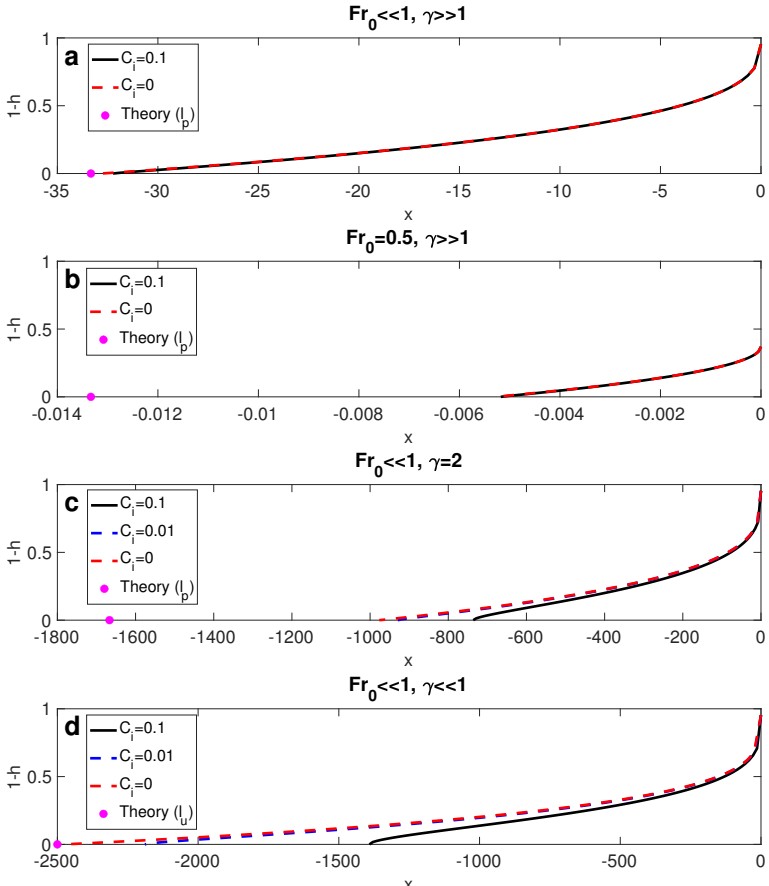

**Figure 2.** Comparison to analytic approximations of seawater intrusion distance, and numerical solutions of shallows water equations under various assumptions. $1 - h$ is the non-dimensional thickness of the seawater layer, which goes to zero at the seawater intrusion distance. (a) Low Froude Number ($Fr_0 << 1$) and densely obstructed macroporous water sheet flow ($\gamma >> 1$). (b) Moderate though sub-critical Froude Number ($Fr_0 = 0.5$) and densely obstructed macroporous water sheet flow ($\gamma >> 1$). (c) Low Froude Number ($Fr_0 << 1$) and moderately (though close to maximum packing density of spheres) obstructed macroporous water sheet flow ($\gamma = 2$). (d) Low Froude Number ($Fr_0 << 1$) and unobstructed water sheet flow ($\gamma << 1$). Other parameters used include: $C_i = 1$ (scaled), and $\Theta = 0$.

orous till layer beneath grounded ice (without significant sheet flow at the ice-till interface). This is in contrast to the layered shallow water flow considered in the previous section where inertial and other terms are important.

The particular problem of seawater intrusion into microporous coastal aquifers where Darcy's law is a valid description of water flow has been extensively studied, due to the important implications for drinking water in coastal areas (for review see

Werner et al., 2013). The canonical case of seawater intrusion into an aquifer considers Darcy flow where freshwater flows from inland at a prescribed rate towards the ocean (illustrated in Fig. 1b). The most well-known method, due jointly to Dupuit (1863) and Forchheimer (1886), starts with the approximation that freshwater flow within an aquifer ($Q_1$) is set by horizontal gradients of the hydraulic head

$$U_1 = -K \frac{\partial \psi}{\partial X} \tag{13}$$

where $K$ is the hydraulic permeability of the microporous medium and $\psi$ is the elevation of the hydraulic head (which is related to the pore pressure within the till). When the aquifer is confined, the right hand side is multiplied by $\frac{H_1}{H}$ to account for $H_1$, the thickness of the freshwater layer, and $H$, the thickness of the till layer. Ghyben (1888) and Herzberg (1901) later showed that when seawater intrudes into an aquifer, the thickness of the buoyant freshwater layer in an aquifer must be hydrostatically compensated by the local thickness of seawater leading to an expression for the freshwater layer thickness

$$H_1 = \alpha(\psi - H_s) - (H_s - H) \tag{14}$$

where $\alpha = \frac{\rho_f}{\rho_s - \rho_f} \approx 40$ is a dimensionless parameter determined by the density difference of freshwater ($\rho_f$) and saltwater ($\rho_s$), and $H_s$ is the sea level relative to the depth of the bottom of the till at the grounding line. The expression above is derived under the assumption that variations in the overburden pressure on the surface of the till layer are small relative to variations in pore pressure due to hydrostatic loading from the ocean. From these two approximations, Strack (1976) showed, that by rearranging equation 14 for $\psi$, taking the spatial derivative of equation 14, and inserting it into equation 13, we arrive at

$$U_1 = -\frac{KH_1}{\alpha H} \frac{\partial H_1}{\partial X}. \tag{15}$$

Integrating this result with respect to $X$ and $H_1$ to find the distance at which the freshwater layer occupies the entire aquifer ($H_1 = H$), we finally arrive at the horizontal extent of seawater intrusion inland in a flat, confined aquifer

$$L_s = \frac{KH}{2\alpha U_{in}} \tag{16}$$

where $L_s$ is the dimensional seawater intrusion distance through soft permeable beds. This solution is widely used in the hydrology community and has been shown to predict observed and laboratory-measured seawater intrusion well (Werner et al., 2013).

On sloped beds, the geometry of the salt-freshwater interface relative to the bed slope becomes a potentially important factor in determining the intrusion distance. Where the bed deepens away from the ocean (referred to as a retrograde or reverse-slope bed in glaciology), the distance of seawater intrusion will be extended as the seawater intrusion flows downhill. For sloped, confined aquifers, equation 14 is

$$H_1 = \alpha(\psi - H_s) - [H_s - H - X \tan(\theta)] \tag{17}$$

where $\theta$ is the bed slope. Lu et al. (2016) extends the approach of Strack (1976), to derive an implicit equation for the horizontal extent of saltwater intrusion

$$H + L_s \tan(\theta_2) + \frac{\alpha H U_{in}}{K \tan(\theta_1)} \ln \left[ 1 + \frac{K \tan(\theta_1) \left( H - L_s \tan(\theta_2) + K \tan(\theta_1) \right)}{\alpha U_{in}} \right] = 0, \tag{18}$$

where $\theta_1$ and $\theta_2$ are the slopes of the lower ice surface and the bed, respectively. For subglacial till layers, we don't expect the lower ice surface slope and bed slope to be significantly different over large areas (which would require the thickness of the till layer to change significantly), so we can safely assume that for our case $\theta = \theta_1 = \theta_2$. This allows us to simplify the above implicit equation, and solve explicitly for the intrusion distance, yielding

$$L_s = -\frac{H}{\tan(\theta)}\left[1 + \frac{\alpha U_{in}}{K\tan(\theta)}\ln\left(1 - \frac{K\tan(\theta)}{\alpha U_{in}}\right)\right] \tag{19}$$

where seawater intrusion occurs if $L_s > 0$. In the next section we will explore the range of predicted seawater intrusion distances from this soft bed theory and the hard bed theory of the previous section.

## 3 Predicted seawater intrusion distance

The theories described in the preceding section can be used to make predictions for the seawater intrusion distance expected over hard and soft beds. In this section we explore the range of seawater intrusion distances that would be predicted to occur, using the wide range of parameter values measured or indirectly inferred from observations of subglacial hydrology near grounding lines and glacier termini. We summarize the range of parameter values we discuss and the corresponding predicted intrusion distances in Table 1.

### 3.1 Hard beds

There are just a few parameters that play a role in determining the horizontal seawater intrusion distance, as can be deduced by examining either the differential equation that exactly specifies the intrusion distance in the model we consider (equation 5) or the analytical approximations on the intrusion distance. In particular the intrusion length scale (equation 12) provides an excellent starting point for understanding which parameters play a role in setting the seawater intrusion distance. We start with the reduced gravity of the two layer system, $g'$, which is generally near 0.27 m/s$^2$ if the fresh and saline layers remain distinct and unmixed (since it depends on the density difference between layers). Compositional variability of either layer may lead to variations in $g'$, but these are likely to be less than 10%. As previously discussed, we assume that entrainment between layers is negligible, though we note here that $g'$ will decrease if entrainment mixes the layers. The drag coefficient, $C_d$, for water flow past ice or obstacles within the subglacial hydrology is typically taken to be between $10^{-3}$ and $10^{-2}$ for a range of observations typically under sea ice (Lu et al., 2011) and lab experiments for idealized walls and objects (Ezhova et al., 2018). We note though, that these prior observations are experiments are for turbulent flow past ice, and laminar flows is likely to have even lower drag coefficients, though exact values are not well-constrained in the literature on flow past ice.

For realistic clast sizes, Hewitt (2011) estimated, using a mathematical model, that the natural thickness scale for a macro-porous subglacial water sheet would be 1-5 cm. Creyts and Schoof (2009) found that the thickness of steady-state subglacial water sheets is set by the size of clasts within the sheet and the water pressure maintained within the system. Antarctic tills are characterized by a wide range of particle sizes (Clarke, 2005), though tills sampled directly from Antarctic ice streams indicate the widespread presence of ploughing boulders up to 10's of cm in size. Consistent with this, MacGregor et al. (2011) esti-

| Case description | $H$ (cm) | $g'$ (m/s$^2$) | $C_d$ | $U_{in}$(cm/s) | $L$ (m) |
|---|---|---|---|---|---|
| Hard, flat bed with thin water sheet | 1 | 0.27 | 0.01 | 1 | 450-700 |
| Hard, flat bed with moderate water sheet | 5 | 0.27 | 0.005 | 0.5 | $\sim 10^4$ |
| Hard, flat bed with thick water sheet | 10 | 0.27 | 0.01 | 0.1 | $\sim 10^5$ |

| Case description | $H$ (cm) | $\alpha$ | $K$ (m/s) | $U_{in}$ (cm/s) | $L$ (m) |
|---|---|---|---|---|---|
| Soft, flat bed with thick, transmissive till | $10^3$ | 40 | $10^{-4}$ | $10^{-4}$ | $\sim 10^1$ |

**Table 1.** Summary of predicted range of intrusion distances discussed in section 3.

mated that a subglacial water sheet of at least 20 cm thickness exists in the grounding zones of Whillans and Kamb ice streams, with the possibility that seawater intrusion explains the persistence of high bed reflectivity across the putative grounding line (discussed further in section 5).

In the same mathematical model discussed above, Hewitt (2011) estimated water velocities in a cm-scale water sheet of 0.5-1 cm/s near the ice sheet margin. Carter and Fricker (2012) modeled water velocities inferred for the Siple Coast from subglacial melt production and subglacial hydropotentials also finding velocities of approximately 1 cm/s. Other studies have estimated subglacial water fluxes from basal melt (Joughin et al., 2009; Pattyn, 2010), with fluxes that are consistent with $\sim 1$ cm/s water velocities through a $\sim$cm thick water sheet or $\sim 0.1$ cm/s through a $\sim 10$ cm thick water sheet ($Re \sim 50$ for both cases, indicating laminar flow). Though a cm-scale water sheet may seem too thin to maintain stratification between layers, a highly stratified salt wedge with low mixing rates ($K_h \sim 10^{-9}$ m$^2$/s is the molecular diffusivity of salt in water, Kunze (2003)) will mix on a time scale of a day, by which time the stratification will have been renewed by freshwater discharge of order cm/s from upstream over a km-scale seawater intrusion. For a $\sim 10$ cm water sheet, mixing of highly-stratified layers would take months. Thus, we consider it completely plausible to maintain distinct layers within cm-scale water sheets if mixing between the layers is slow. We also note that even at more moderate levels of mixing ($K_h \sim 10^{-7}$ m$^2$/s due to, e.g., double diffusive convection, van der Boog et al. (2021)), it is still possible to maintain two distinct layers in water sheets $\sim 10$ cm thick.

For subglacial macroporous water sheets, $\gamma = 2\phi H/\pi d(1-\phi)$ is dependent on the density and size of obstacles in the macroporous subglacial water sheet. We expect that the thickness of the sheet ($H$) will, at most, be the typical diameter of clastic obstacles ($d$), which maintain the ice-bed opening. Spherical objects of nearly similar size have a maximum 3-D packing density of $\phi = \pi/\sqrt{18} \approx 0.74$ (from the "Kepler Conjecture"; Hales, 2005). Consequently, we expect that the maximum value of $\gamma$ is likely to be near $2/(\sqrt{18} - \pi) \approx 2$. This argument provides a reasonable upper bound on $\gamma$, though it should be kept in mind that nature does not necessarily arrange clasts in an optimal packing configuration and that clasts may in reality vary in size.

Using this range of values for the thinnest water sheet ($H = 1$ cm), the highest drag for a wall with less than cm-scale roughness($C_d = 0.01$), and the highest water velocity ($U_{in} = 1$ cm/s), we calculate (from equation 12) a hard bed intrusion length scale of $\tilde{L} = 2700$ meters, giving intrusion distance of 450 meters for a densely obstructed water sheet or 700 meters for an unobstructed water sheet (from the lower and upper bounds in equation 11, respectively). A more moderate range of parameters ($H = 5$ cm, $C_d = 0.005$, $U_{in} = 5$ mm/s) gives intrusion distances in the range of tens of kilometers. At the other end

of the spectrum, a thick water sheet suggested by radar to exist at Siple Coast grounding zones ($H = 20$ cm; MacGregor et al., 2011), with drag coefficients appropriate for large roughness or dense clasts ($C_d = 0.01$), and low water velocity ($U_{in} = 1$ mm/s), would suggest an intrusion length scale easily spanning the entire ice sheet interior. Maintaining steady water sheet thickness in this range would require very high water pressures which may lead to turbulent flow, violating one of the base assumptions of our theory ($Fr < 1$). We also would not expect such conditions to persist all the way into the ice sheet interior (where subglacial water discharge and water sheet thickness are both likely to be much lower, in addition to potential variations in bed slope). Therefore, we note this possibility, but remain skeptical that our theory would provide a reliable prediction for seawater intrusion distance in this case. We should also note there have been indirect observations of very thin water sheets of ~mm thick (Engelhardt and Kamb, 1997) at ice-till interfaces in Antarctica, but we do not envision that maintaining distinct layering would be possible in such a circumstance.

This extreme range in potential intrusion distance reflects the fact that subglacial hydrological parameters are uncertain over orders of magnitude, resulting in many orders of magnitude uncertainty in the intrusion distance. Nonetheless, we conclude that on hard beds, seawater intrusion distances of at least 100's of meters are possible even under very conservative assumptions, and intrusion distances of 10's of kilometers are plausible. In section 4, we explore the implications of intrusion-induced basal melt over such length scales upstream of marine ice sheet grounding lines.

## 3.2 Soft beds

When subglacial drainage occurs by Darcy flow through confined layers of till (i.e. a "soft bed"), there is a distinct (from the hard bedded case discussed above) horizontal length scale governing seawater intrusion, given by equation 16. The $\alpha$ parameter (analogous to $g'$) is fairly well known for freshwater-seawater systems to be 40. $H$ is the thickness of the confined till layer, which can range anywhere from cm to tens of meters. Similarly, the inflow rate of freshwater through till layers ($U_{in}$) has been estimated to be on the order of $0.1 - 1$ mm/s in ice streams where frictional heating at the ice-till interface generates high melt rates (Joughin et al., 2009). However, in regions of lower basal melt rates, we might expect inflow rates closer to $0.001 - 0.01$ mm/s (Pattyn, 2010).

Though $H$ and $U_{in}$ are uncertain over 1-4 orders of magnitude, by far the most uncertain property of subglacial till relevant to seawater intrusion is $K$, the hydraulic conductivity of till. Laboratory measurement of $K$ in till from formerly and currently glaciated field sites ranges from $10^{-12}$ to $2 \times 10^{-6}$ m/s (Freeze and Cherry, 1979; Cuffey and Paterson, 2010). Other methods have also been used to measure till hydraulic conductivity including: in-situ measurement of borehole fluids far upstream of the grounding line at West Antarctic ice streams (Engelhardt et al., 1990) and inference from measurements of grounding line migration in different West Antarctic ice streams (Warburton et al., 2020). These other approaches to constraining till properties give a range of $10^{-9}$ to $10^{-4}$ m/s for till hydraulic conductivity.

Even though the range on the till hydraulic conductivity, thickness and inflow discharge velocities is large, we can confidently conclude that, based on the scaling given in equation 16, seawater intrusion into flat, confined till layers will extend, at most, meters upstream of the grounding line. For most of the range of till properties discussed above, the length scale of seawater intrusion on flat beds will be negligible. This can be compared to the range of seawater intrusion into coastal aquifers of

320 meters to kilometers. Hydraulic conductivity of consolidated subglacial till is well known to be low compared to a typical aquifer, which is largely the reason for such small intrusion distance through flat soft beds. It is however, important to note, that previous field-based studies of till properties (particularly those reporting higher hydraulic conductivities) suggest that saturation of the subglacial till layer may prevent infiltration of basal melt below the ice-till interface, leading to sheet-like flow between ice and till, more similar to the hard-bed sheet-like hydrology discussed in sections 2.1 and 3.1 (though the thickness

of such a supra-till water sheet is not entirely clear). Thus, as observations from West Antarctica suggest (MacGregor et al., 2011; Horgan et al., 2013), even in places where there is substantial evidence for the existence of thick subglacial till layers, seawater intrusion may still occur through shallow water flow between ice and till.

## 3.3 Sloping beds

The intrusion of seawater under grounded ice is ultimately driven by gravity. The salt wedge is the thickest near the ocean and

330 thins towards the glacier interior (Fig. 1), producing a pressure gradient (proportional to the salt wedge thickness gradient, $\frac{\partial h}{\partial x}$ in equation 5) that maintains the salt wedge against resistance from drag at walls, obstacles and the layer interface. On flat beds, the layer interface becomes flatter towards the interior (e.g., Fig. 2) such that eventually the drag exceeds gravitational driving, causing the termination of seawater intrusion. A bed that deepens into the glacier interior ($\theta > 0$ in equation 4 or equation 19) slopes in the same direction as the salt wedge interface, and thus drives further intrusion over what would be expected on a

335 flat bed. A sufficiently reverse-sloping bed will drive additional and potentially unbounded seawater intrusion distance (i.e., $L \to \infty$). Here we explore the criteria for the "critical bed slope" beyond which seawater intrusion extends unbounded for both hard and soft beds. We will also numerically solve for the intrusion distance over a range of bed slopes and other parameter values to determine generally when bed slope becomes an important driver of intrusion.

Following Wilson et al. (2020), we note that where the right hand side of equation 5 becomes less than or equal to zero, the

340 drag terms will never be sufficient to overcome the gravitational driving of the seawater intrusion, implying a seawater layer with constant or growing thickness (towards the glacier interior). This implies a "critical bed slope" criterion

$$\theta > \eta C_0 F r_0^2 \tag{20}$$

where $\eta$ is a dimensionless number that is the minimum of the function $\widetilde{C}_i(1-h)^{-1} + \widetilde{C}_d(1+\gamma h)$ and $C_0$ is the drag coefficient scale (which is set by the obstacle/wall drag, $C_d$). Analytical approximation of $\eta$ generally yields unwieldy expressions in terms

of $\widetilde{C}_i$, $\widetilde{C}_d$ and $\gamma$. However, we can make two general observations about the scales of $\eta$ before turning to numerical solution. For an unobstructed water sheet ($\gamma << 1$), $\eta$ is generally $O(1)$. For a macroporous water sheet ($\gamma \sim O(1)$), $\eta$ is generally $O(\gamma)$. Thus, for most realistic configurations of the subglacial water sheet ($\gamma \lesssim$), we expect the critical bed slope to be $O(C_0 F r_0^2)$.

To show how bed slope affects seawater intrusion distance generally, we numerically solve equation 5, and in Fig. 3 plot the seawater intrusion distance ($L$) over a wide range of bed slopes and Froude numbers. What we see largely reflects the inequality

in equation 20. For realistically low flow rates of subglacial discharge ($Fr_0 \sim O(0.1)$) and drag coefficients ($C_0 \sim 5 \times 10^{-3}$), seawater intrusion becomes unbounded when the bed has a reverse slope that is steeper than $1-5 \times 10^{-4}$. Additionally, prograde bed slopes that are common ($\theta < 10^{-3}$) effectively rule out seawater intrusion of more than 10 meters. We generally see that,

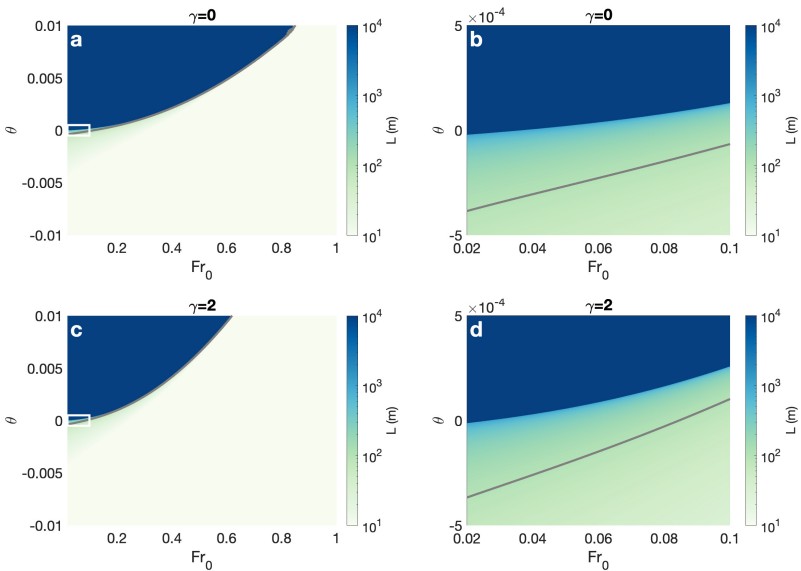

**Figure 3.** Seawater intrusion distance over a hard bed, calculated numerically from full model (equation 5) over a range of bed slope and Froude Number. For illustrative purposes, the bed slope ($\theta \approx \Theta C_0$) and intrusion distance ($L = \ell H / C_0$) have been shown in dimensional form for moderate parameter values ($C_0 = 0.005$ and $H = 5$ cm). (a-b) Unobstructed water sheet ($\gamma = 0$). White box in panel (a) indicates region magnified in panel (b). (c-d) Obstructed macroporous water sheet ($\gamma = 2$). White box in panel (c) indicates region magnified in panel (d). In all panels, $C_d / C_i = 10$. Blue regions are generally above the critical slope. Grey contour indicates 100 m seawater intrusion distance.

for subcritical water flow ($Fr_0 < 1$) over reverse bed slopes, the critical bed slope is $O(10^{-2})$ or flatter. These bed slopes are well within the range of reverse bed slopes found at many grounding lines at glacier termini in Greenland and Antarctica (Ross et al., 2012; Morlighem et al., 2017, 2020).

For a sloped subglacial porous aquifer, there is a similar limit in which the intrusion length becomes unbounded. This occurs when the slope of the seawater intrusion interface is less than the bed slope, thus meaning that they will never make contact (unless the bed slope or freshwater inflow changes). This occurs when the logarithm in equation 19 becomes unbounded, which is the case for

$$\theta > \frac{\alpha U_{in}}{K} \tag{21}$$

assuming $\theta \approx \tan(\theta)$. $\frac{\alpha U_{in}}{K}$ is a dimensionless parameter that quantifies the relative importance of freshwater discharge and hydraulic conductivity of the till, which together determine the "resistivity" of the till to seawater intrusion. When this parameter is large, seawater will have a difficult time intruding into the till either due to large freshwater discharge pushing back, and/or low hydraulic conductivity preventing easy flow. At the upper bound of the range of observed till hydraulic conductivity: $K \in [10^{-11}, 10^{-4}]$ m/s and the lowest feasible freshwater flow rates in till from the interior of the catchment ($U_{in} \sim 10^{-6}$), this "resistivity" is $O(0.1)$ and the critical bed slope is in the range of the absolute steepest reverse bed slopes in either Greenland

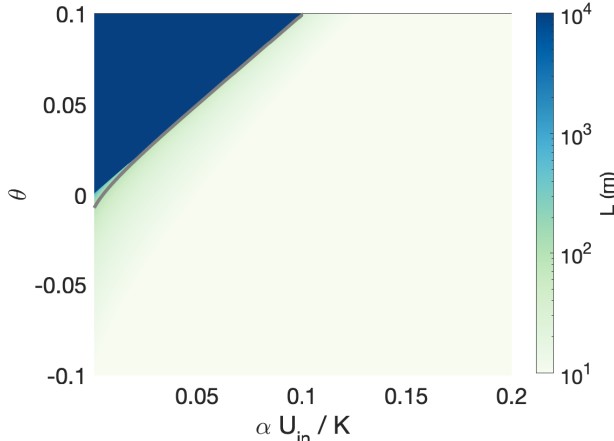

**Figure 4.** Seawater intrusion distance through a soft bed, calculated numerically from equation 19 over a range of bed slope and $\alpha U_{in}/K$ (a dimensionless parameter indicating the relative strength of freshwater discharge to hydraulic conductivity). Grey contour indicates 100 m seawater intrusion distance.

or Antarctica. Indeed, calculating seawater intrusion distance from equation 19 for a wide range of $\theta$ (Figure 4) shows that while there are some circumstances under which seawater intrusion into till aquifers might be non-negligible (i.e. greater than meters), such conditions are seemingly rarely attained. However, it is important to emphasize that if the hydraulic conductivity of till was observed to be any higher than $10^{-4}$ m/s or freshwater discharge was very low (perhaps under a slowly flowing glacier), seawater intrusion into till would become a distinctly non-negligible possibility on reverse-sloping beds.

## 4 Melt from seawater intrusion beneath grounded ice sheets

Where water comes into contact with ice sheets, heat and salt are exchanged across the ice-water boundary layer (Jenkins, 1991), which may lead to dissolution or melting of ice. To estimate the effect of such intrusion-induced basal melt of grounded ice on the evolution of marine ice sheets, we incorporate a simple parameterization of intrusion melt into the Ice Sheet and Sea Level System Model (ISSM; Larour et al., 2012). The parameterization assumes that basal melt rates ($\dot{m}$) decrease linearly from the grounding line upstream, reaching zero at a specified intrusion distance, $L$

$$\dot{m}(x) = \dot{m}_{GL}\left[1 - \frac{x}{L}\right] \tag{22}$$

where $\dot{m}_{GL}$ is the melt rate at the grounding line (either parameterized or determined via coupling to an ocean model). In this parameterization in ISSM, $x$ is the horizontal distance to the nearest grounding line. In an ice sheet model with explicitly simulated subglacial hydrology, $x$ could be taken as a horizontal distance along hydropotential minima. We also specify that this parameterization is only applied upstream of the grounding line, where $0 < x < L$. Figure 5 shows an illustration of this parameterized melt rate as a function of distance from the terminus or grounding line.

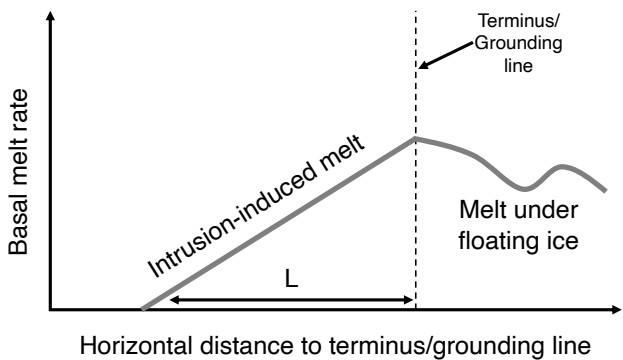

**Figure 5.** Illustrative schematic of a parameterization for basal melt under grounded ice caused by seawater intrusion.

Observations show that there are many places where a cold, fresh layer exists between floating ice and the warm, salty
seawater. In such circumstances, basal melt still occurs through double-diffusive convection (Kimura et al., 2015; Begeman
et al., 2018), though it is lower than it would be if a fully mixed and turbulent boundary layer existed at the ice-ocean interface
(Rosevear et al., 2021). Where there is intrusion of a warm, saline layer under fresh, cold subglacial discharge beneath grounded
ice, with subcritical water flow and limited entrainment (as may exist for seawater intrusion over hard beds), we argue that
double-diffusive convection may occur and maintain distinct water layers while driving some ice melt that is dependent on
properties of the saline layer, as has been shown in experiments and direct numerical simulations of the sub-ice boundary layers
(Martin and Kauffman, 1977; Turner and Veronis, 2004). Correspondingly, in our first model configuration, we consider two
sets of simulations. In the first set of simulations, grounding line melt rates are 10's of m/yr, in line with previous benchmark
simulations (described in the next section). In a second set of simulations, grounding line melt rates are m/yr, similar to those
observed in a thin (i.e. 10's of cm) double-diffusive staircase beneath the George VI ice shelf (Kimura et al., 2015). It is
important to note two other possible outcomes from such a strongly stratified subglacial water layer: (1) where the sub-ice
fresh layer is very thick (i.e. meters), double-diffusive convection drives even lower basal melt rates (Begeman et al., 2018),
and (2) where seawater temperature is below the freshwater freezing point, there will be freeze-on at the the ice base, not
melting (Martin and Kauffman, 1977; Pedley et al., 1988). Rather, the goal of this section is to explore the consequences
of intrusion-induced melt where the freshwater layer is sufficiently thin and the seawater layer is sufficiently warm, which
currently occurs at some locations in Greenland and Antarctica, and may occur at more locations in the future.

Intrusion-induced basal melt fluxes would need to be balanced by an additional influx of seawater to maintain steady-state
seawater intrusion distance (and layer thicknesses). For intrusion-induced basal melt rates of the form of equation 22, the
integrated melt flux would be $\dot{m}_{GL}L/2$. From the grounding line boundary condition on freshwater flow speed over a hard
bed (equation 7), we can deduce the dimensional saline layer thickness at the grounding line: $H_2(x=0) = H(1 - Fr_0^{2/3})$.
Ocean currents can cause an incoming flux of $U_o H(1 - Fr_0^{2/3})$ where $U_o$ is the incoming current velocity from the ocean.
For intrusion distances of hundreds to thousands of meters, subglacial water layers thicknesses of 1-10 cm, and freshwater

discharge velocities of 0.1-1 cm/s, $U_o$ needs to be 0.1-10 cm/s to balance basal melt rates of order 1-100 m/yr. This is well within the range of current speeds measured near the grounding lines of ice streams in Antarctica (Begeman et al., 2018). Thus, we consider these melt rates to be plausible purely from a consideration of the conditions necessary to maintain a steady-state.

However, we do note that such considerations do limit the possible range of steady-state intrusion distances and basal melt rates (i.e., intrusive melt rates greater than 100 m/yr over 10's of km would only be possible to maintain in a steady-state with very rapid ocean inflow to compensate). Future work should explore the internal circulation of steady-state seawater intrusion layers in more detail.

Our goal in this section is to implement a simple parameterization of intrusive melt to estimate the effect of intrusion-driven 415    melt under grounded ice on marine ice sheet evolution in two well-studied model configurations. However, we do recognize that the parameterized linear decrease in melt rates in equation 22 is clearly a simplification of the shape of the warm salt wedges shown in Fig. 2 and does not consider the potential complexities of sub-ice boundary layer processes. In reality, a more realistic fluid dynamic model including turbulence and heat fluxes would be needed to fully understand melting in seawater intrusions. In section 5, we discuss the myriad ways that the representation of intrusion-driven melt can be made more realistic 420    in numerical models.

In this study, we have adapted the "sub-element melt 2" (SEM2) parameterization described by Seroussi and Morlighem (2018) to incorporate melt from seawater intrusion on grounded ice by calculating the "level set" horizontal distance from the grounding line everywhere in the ISSM domain ($x$ in equation 22). Intrusion distance $L$ is specified as a model parameter and sub-shelf melt rates $\dot{m}_{GL}$ is prescribed as a constant over floating ice (though it could equally well come from a more 425    sophisticated model for floating melt rates). In this section, we will explore the effect of different intrusion distances on transient evolution of a marine-terminating terminating glacier in two benchmark model configurations: the MISMIP+ idealized bed topography (Asay-Davis et al., 2016) and future evolution of Thwaites Glacier in West Antarctica (based on a well-tested model configuration; Seroussi et al., 2017; Robel et al., 2019). All simulations in this section are conducted at 125 meter horizontal resolution, where errors due to the precise form of the numerical implementation of basal melt near the grounding 430    line are negligible, as shown in Seroussi and Morlighem (2018). A convergence study (not plotted) indicates that all results presented here using the intrusion melt parameterization are less than 3% different from equivalent simulations conducted at 250 meter horizontal resolution, confirming that these results are indeed converged in horizontal resolution.

## 4.1   ISSM Intrusion Melt: MISMIP bed topography

To demonstrate the effect of melt from seawater intrusion on ice sheet mass loss, we start by using a common benchmark of 435    marine ice sheet models, the MISMIP+ model configuration (Asay-Davis et al., 2016; Cornford et al., 2020). In this idealized configuration, a marine-terminating glacier with a buttressing ice shelf begins in a steady-state with the grounding line on a reverse-sloping bed, with no basal melt applied to the floating ice, and constant snowfall accumulation rate in space. For simplicity, in this study we consider a variant on the "Ice1r" transient experiment, in which we apply a constant melt rate on all floating ice, and then permit melt to occur with a linearly decreasing profile upstream of the grounding line according to 440    our intrusion melt parameterization (equation 22 and Fig. 5). This constant-melt variant ensures that any dynamic feedback in

response to melt forcing is due to ice sheet dynamics alone and not complicated feedbacks due to melt rate dependencies on ice sheet and bed geometry. This simplification will also allow us to straightforwardly compare the effect of adding intrusion melt over certain distances to cases without intrusion melt (in the discussion below).

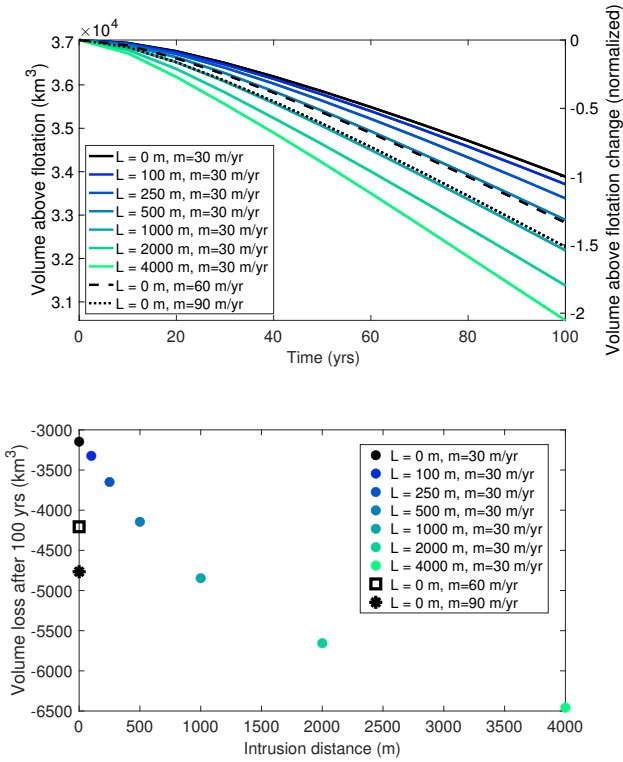

**Figure 6.** ISSM simulations of transient ice volume loss in the MISMIP+ configuration over 100 years with varying distances of "high" ($\dot{m}_{GL} = 30$ m/yr) intrusion-induced basal melt upstream of the grounding line and increased submarine melt rates with no intrusion for comparison. (a) Transient ice volume loss over 100 years. Color variation shows different prescribed intrusion distances, dashed and dotted lines show higher submarine melt rates with no intrusion. (b) Total ice volume loss at the end of 100 years as a function of intrusion distance.

In Fig. 6, we show the transient ice loss (in terms of volume above floatation, VAF) in the MISMIP+ configuration in response to various seawater intrusion distances ($L$ in equation 22) and high prescribed floating melt rates ($\dot{m}_{GL} = 30$ m/yr in equation 22). The solid colored lines (in panel a) and colored circles (in panel b) show ice loss over 100 years. In this "high melt" scenario, we find a robust and significant effect of melt from seawater intrusion, with a 10-50% increase in the rate of ice volume loss for seawater intrusion over hundreds of meters, and a 50-105% increase in loss rate for intrusion over kilometers. The direct increase in ice loss from basal melt in the intrusion region generally amounts to less than 5% of the increased volume loss across experiments. Rather, it is the dynamic marine ice sheet response to melting upstream of the grounding line which is responsible for the significant increase in ice loss in the experiments with intrusion-induced basal melt. Intrusion melt turns

grounded ice into floating ice, which is then subject to higher melt rates and induces a greater flux of ice from upstream. The effect of 500 m of seawater intrusion is equivalent to doubling the floating melt rate without intrusion (dashed line), and the effect of 1 km of seawater intrusion is equivalent to tripling the floating melt rate without intrusion (dotted line).

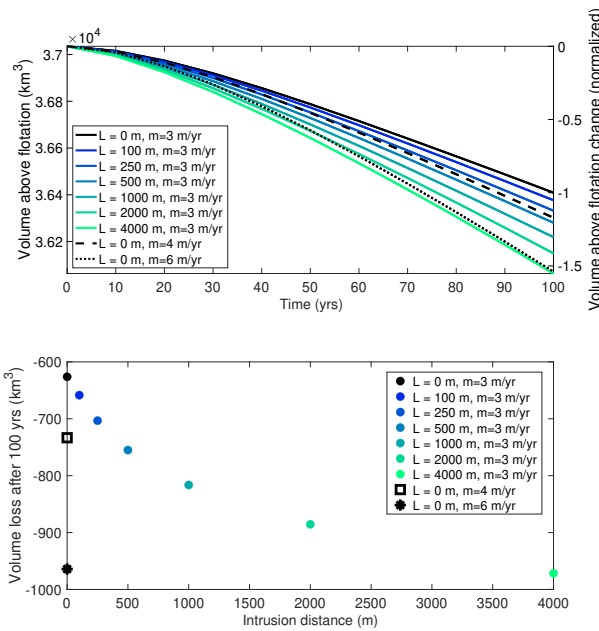

**Figure 7.** ISSM simulations of transient ice volume loss in the MISMIP+ configuration over 100 years with varying distances of "low" ($\dot{m}_{GL} = 3$ m/yr) intrusion-induced basal melt upstream of the grounding line and increased submarine melt rates with no intrusion for comparison. (a) Transient ice volume loss over 100 years. Color variation shows different prescribed intrusion distances, dashed and dotted lines show higher submarine melt rates with no intrusion. (b) Total ice volume loss at the end of 100 years as a function of intrusion distance.

As noted previously, it may be difficult to maintain high basal melt rates under grounded ice in steady-state. So, we also conduct comparable "low melt" simulations with an order of magnitude lower baseline basal melt rates ($\dot{m}_{GL} = 3$ m/yr), which is closer to the melt rates expected in locations with double-diffusive convection driving melt (Kimura et al., 2015). In Fig. 7, we find a 5-20% increase in the rate of ice volume loss for seawater intrusion over hundreds of meters, and a 20-55% increase in loss rate for intrusion over kilometers. The effect of 500-1000 m of seawater intrusion is equivalent to increasing the floating

melt rate without intrusion by 1/3 (dashed line), and the effect of 4 km of seawater intrusion is equivalent to doubling the floating melt rate without intrusion (dotted line). Thus, while these intrusion-induced increases in ice loss rate for the "low-melt" scenario are less than in the "high-melt" scenario, intrusion does still have a first-order effect on ice loss even for basal melt rates more typical of double-diffusive convection.

     Though these simulations are instructive as to the large effect of intrusion-induced melt, we do emphasize that the exact sen-

sitivity of ice loss to prescribed floating ice melt rate and intrusion distance is dependent on the particular model configuration

(i.e. bed topography, extent of buttressing, initial glacier state, etc.). In the next section, we test another model configuration of interest to provide a non-idealized configuration as a point of comparison.

## 4.2 ISSM Intrusion Melt: Thwaites Glacier, West Antarctica

We also test effect of melt from seawater intrusion on ice loss from Thwaites Glacier (TG), a marine-terminating glacier in
West Antarctica that is the focus of intense interest due to its recent acceleration and contribution to global sea level rise. The extent and intensity of submarine melting under the floating portions of TG have been studied extensively using field and remote sensing observations (Bevan et al., 2021; Wåhlin et al., 2021, e.g.,), and ocean modeling (Seroussi et al., 2017; Nakayama et al., 2019, e.g.,). In this section, we adapt the model configuration of ISSM from Seroussi et al. (2017), with a domain encompassing the TG catchment and a fine horizontal resolution. Submarine melt rates are again set to be constant
at 60 m/yr which promotes a rapid retreat of the TG grounding line and is consistent with observations of Thwaites sub-shelf melt rates (Milillo et al., 2019). Surface mass balance is held constant in time and is based on regional simulations with RACMO2 averaged over the 1979-2010 time period (Lenaerts et al., 2012). The characteristics of subglacial hydrology of TG have been the focus of many recent studies, with evidence for a soft bed of variable strength, and channelized hydrology near the grounding line (Schroeder et al., 2013, 2016) which sits on a steeply reverse sloping bed (Morlighem et al., 2020) that,
in places, exceeds the critical bed slope for unbounded seawater intrusion discussed in section 3.3. These characteristics of the subglacial environment are generally favorable for the intrusion of seawater, even in particular channelized regions, which is consistent with the observations of localized high basal melt upstream of the TG grounding line by Milillo et al. (2019). However, observations of TG subglacial hydrology are still sufficiently uncertain that we simply explore the potential effect of intrusion-induced melt upstream of the grounding line for the evolution of TG (similarly to the MISMIP+ case).

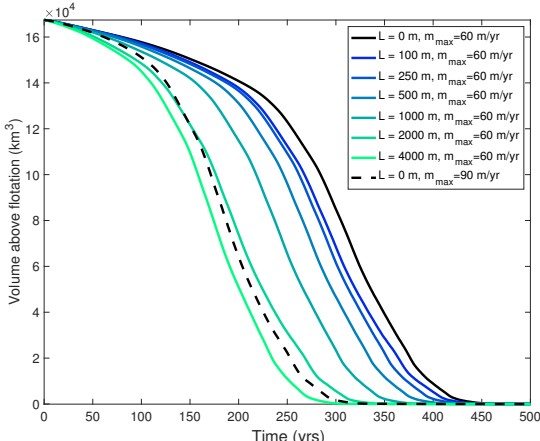

**Figure 8.** ISSM simulations of transient ice volume loss of Thwaites Glacier over 500 years, with varying distances of intrusion-induced basal melt upstream of the grounding line (solid colored lines) and with increased submarine melt rate with no intrusion (black dashed/dotted lines) for comparison.

In Fig. 8, we show the transient ice loss over 500-year simulations in which TG retreats completely through its catchment (in all simulations). Seawater intrusion distances of hundreds of meters accelerate the onset of the most rapid retreat by decades, and seawater intrusion over kilometers accelerates the onset of the most rapid retreat by 100-200 years. In the first 100 years of TG retreat (a common simulation period), hundreds of meters of seawater intrusion increase the rate of ice loss at TG by up to 20%, while kilometers of seawater intrusion increase the rate of ice loss by 20-100%. The effect of 2-4 km of seawater

intrusion is equivalent to increasing the submarine melt rate by 50% without intrusion (dashed line). Though there are some quantitative differences between TG and MISMIP+ simulations, in the sensitivity of ice loss rate to seawater intrusion, it is generally the case that seawater intrusion of hundreds to thousands of meters (well within the range of feasibility as shown in section 3) leads to substantially more ice loss from retreating marine-terminating glaciers. These results suggest that melt from seawater intrusion could be a very important process in driving ice loss from marine ice sheets. In the next section, we discuss

the implications of these results for observing the ice-ocean interface and simulating intrusion in ice sheet models.

## 5   Discussion

Future evolution of marine ice sheets is strongly sensitive to the potential for seawater intrusion under grounded ice. However, as we have shown, predictions of the horizontal extent of seawater intrusion remain uncertain over many orders of magnitude, primarily due to the lack of constraints on subglacial hydrology near the ice-ocean interface. To make better predictions of

how seawater intrusion may affect future marine ice sheet evolution will require: more sophisticated numerical models of the ice-ocean interface, and more observational and experimental constraints on the properties of subglacial hydrology near the ice-ocean interface. In particular, future work can focus on answering three questions: (1) can we definitively confirm that seawater intrusion and melt are happening under grounded ice at some glaciers? (2) what are the properties of the subglacial hydrologic system under grounded ice near the ice-ocean interface? (3) which processes play a role in the fluid dynamics of

the freshwater-seawater interface?

### 5.1   "Effective" intrusion melt in large-scale ice sheet models

We have shown that the extent of potential seawater intrusion and melt under grounded ice are highly consequential for the future evolution of marine ice sheets. This is consistent with the conclusion of previous studies that have found that grounding line migration is most sensitive to basal melt at the grounding line (Parizek et al., 2013; Arthern and Williams, 2017; Reese

et al., 2018; Goldberg et al., 2019). Seroussi and Morlighem (2018) further showed that, in some ice sheet models, basal melt intended for floating ice is "erroneously" applied to areas just upstream of the grounding line. Such misapplication of basal melt will lead to more rapid marine ice sheet retreat and ice loss (e.g., by up to 100% compared to cases without such "errors"; Seroussi et al., 2019). Other studies of past and future ice sheet evolution due to ocean warming (e.g., Golledge et al., 2017) have used such a coarse-resolution ice sheet model in which ocean melt is applied to partially grounded elements, effectively

including many kilometers of seawater intrusion in their simulations. Such models simulate collapse of the West Antarctic Ice Sheet and portions of the East Antarctic Ice Sheet during the Pliocene warm period, explaining the much higher global sea level

during this period that had been a challenging target for other ice sheet models (Dutton et al., 2015). Such models have also been used to simulate the future contribution of the Antarctic Ice Sheet to sea level rise (Golledge et al., 2019), falling near the high end of the range of future model projections (Seroussi et al., 2020). Indeed, ice sheet intercomparisons have demonstrated

that marine ice sheet models which include such "effective" intrusion-indued basal melt on grounded ice are consistently more sensitive to ocean forcing than models in which no melt is applied to these elements (increasing the Antarctic contribution to sea level rise by 50-100%; Seroussi et al., 2019). Other ice sheet dynamical mechanisms (particularly those controlling the rate of ice loss from the ice-ocean interface) have also been incorporated into models as a way of explaining the higher marine ice sheet sensitivity to climate change implied by Pliocene sea level (e.g., DeConto and Pollard, 2016). Seawater intrusion

would appear to have a similar degree of support as these other sensitivity-boosting mechanisms both from fundamental theory and contemporary glaciological observations. The potential role of seawater intrusion in explaining past high sea levels and substantially increasing future projections of sea level rise suggests that this process deserves more attention through observational and modeling efforts to constrain and predict subglacial hydrology near the ice-ocean interface.

## 5.2  Models of seawater intrusion

In this study, we showed that intrusion distance is sensitively dependent on the bed slope and subglacial hydrology. A more sophisticated treatment of seawater intrusion in an ice sheet model would dynamically calculate $L$ at all grounding lines or termini, depending on the local bed type (hard or soft), bed slope, and state of subglacial hydrology. Unfortunately, the extent of soft and hard beds is still not well-known below the Greenland and Antarctic ice sheets, most marine ice sheet models do not dynamically simulate subglacial hydrology, and those that do differ widely in the assumptions and equations used to describe

evolving drainage (de Fleurian et al., 2018).

The theory in this study also makes many assumptions in order to explore the generic importance and extent of seawater intrusion. One assumption is that the two laminar layers experience drag at their interface and with the surrounding solid boundaries (ice, bed, clasts or till grains). Another important assumption is that subglacial discharge is subcritical (i.e. $Fr > 1$), which permits the existence of layered subglacial flow. Though we posit that seawater intrusion could produce basal melt

via a process such as double-diffusive convection, we do not model the sub-ice boundary layer in detail here. Additionally, compositional heterogeneities and flow from the ocean could further complicate our simplified assumptions of a uniformly salty body of seawater at rest. Transient evolution of the salt wedge from tidal pumping (as in Walker et al., 2013; Sayag and Worster, 2013) or evolution of subglacial hydrology through melting by seawater may also complicate matters. Lateral heterogeneities in subglacial hydrology and other bed properties (e.g., channelization, bed slope) may lead to intrusive melt

at some parts of the ice-ocean interface but not others. As we show, the sensitivity of ice sheet evolution to intrusion-induced melt is strongly sensitive to the extent of seawater intrusion and heat fluxes from the intrusion into grounded basal ice. The potentially substantial importance of seawater intrusion in determining the response of marine ice sheets to ocean warming necessitates detailed considerations of all the complexities listed above. Future studies should utilize high-resolution two- and three-dimensional numerical modeling of the fluid and thermal transition between subglacial water flow and the ocean

circulation, potentially coupled to ice sheet models. This study should provide the motivation for future model developments in this direction.

## 5.3 Observations and experiments on seawater intrusion

Prior studies have found intriguing evidence of seawater intrusion in observations of grounding lines. MacGregor et al. (2011) observe no discernable change in the bed reflectivity in radar transects across the grounding lines at Whillans and Kamb
ice streams, indicating a continuous water layer without a strong change in composition across this transition region (up to 10's of kilometers upstream). Though they ultimately attribute these observations to other factors (leaching from subglacial sediments), they do consider the possibility that seawater intrusion may explain these observations (though ultimately dismiss this possibility due to the subglacial hydropotential gradient argument, discussed in section 2.1). Horgan et al. (2013) and Drews et al. (2017) both observe subglacial channels across grounding lines in West and East Antarctica, and both argue that
weak hydraulic potentials would suggest the potential for tidal pumping of seawater upstream of the grounding line as in an estuary. Such an observation is fully consistent with the independent observations of MacGregor et al. (2011), and the "estuary-like" salt wedge described here and previously for channels in Wilson et al. (2020), though the theory does not require tides for seawater to intrude past the grounding line. Milillo et al. (2019) observed subglacial melt from satellite-based Interferometric Synthetic Aperture Radar kilometers upstream of the grounding line along channelized pathways. Though catchment-spanning
seawater intrusion is likely ruled out by the lack of seawater observed in boreholes drilled to the bed far upstream (100's of km) of the grounding line (Tulaczyk et al., 2014), there are indications of high-salinity water far upstream of the grounding line in recent electromagnetic observations (Gustafson, 2020).

Interferometry, sounding radar, and seismic methods provide complementary observations of the ice-bed interface (e.g., bed reflectivity, melt rate, acoustic impedance). More recent advances in radar technology and processing (e.g., ApRES, full-
570 waveform inversion) and electromagnetic methods (Key and Siegfried, 2017) hold promise to better constrain the structure of subglacial water system (e.g., $H$), pore-water properties, the salinity of subglacial water, and basal melt rates at high resolution across the grounding line. Providing better constraints on the velocity of subglacial water flow from upstream ($U_{in}$) in either till or subglacial water sheets would likely require either direct access through borehole drilling near the grounding line (as in Engelhardt and Kamb, 1997, but closer to the grounding line) or model-based estimates of meltwater production upstream
(Joughin et al., 2009; Pattyn, 2010; Carter and Fricker, 2012). Additionally, drag coefficients for a wider range of subglacial conditions and obstacle types can be better constrained with experimental techniques, though only limited efforts have been made in this direction (Prohaska, 2017). Ultimately, targeted efforts to measure seawater intrusion in the field and in laboratory experiments would provide important constraints on the theory developed in this study.

## 6 Conclusions

The interface of ice, ocean, and the solid Earth plays host to a complex array of processes that drive much of the changes observed at marine ice sheets. This study considers one such process, the intrusion of seawater under grounded ice. We extend

the theory of layered seawater intrusion under grounded ice developed for channels by Wilson et al. (2020), to a generalized theory for seawater intrusion under grounded ice in many possible subglacial hydrological systems. We find that there is a wide range of seawater intrusion distances predicted by this generalized theory, which includes cases with effectively no intrusion, but also up to tens of kilometers of intrusion (and potentially even further) in macroporous water sheets (or channels) over impermeable beds. Seawater intrusion is generally negligible through microporous till layers, but under the right circumstances (steep reverse bed slopes, high hydraulic conductivity) till can support substantial intrusion, analogous to seawater intrusion in porous coastal aquifers. Critically, if seawater intrusion melts grounded ice, there can be a substantial increase in the rate of transient ice loss from marine ice sheets. This finding is in line with previous studies which have found strong sensitivity of the projected contribution of marine ice sheets to future sea level rise to the numerical treatment of basal melt just upstream of grounding lines in ice sheet models (Seroussi et al., 2019).

This study demonstrates that seawater intrusion under grounded ice is theoretically possible (even expected) and has substantial implications for projections of future sea level rise from marine ice sheet retreat under climate change. To determine whether seawater intrusion actually occurs in reality, targeted observational campaigns and experiments are needed to investigate the subglacial hydrology and melt rates near grounding lines and glacier termini. Additionally, the sensitivity of sea level projections to uncertainties in intrusion-induced melt should be tested in a deliberate fashion (rather than as a numerical artifact). High resolution experimental and numerical approaches are also needed to understand the potential role of turbulence, tides and other important processes in either supporting or suppressing seawater intrusion. Though seawater intrusion has thus far been understudied as a potential driver of ice sheet changes, we hope that synthesizing theoretical and modeling approaches to intrusion will catalyze future efforts to better under this elusive process.

*Code availability.* All Python and MATLAB scripts used to produce the figures in this manuscript are freely available in a public GitHub repository: https://github.com/aarobel/IntrusionUnderIceSheets. The results of sections 2-3 can also be reproduced interactively in the cloud using the binderized Jupyter notebook: https://mybinder.org/v2/gh/aarobel/IntrusionUnderIceSheets/master?filepath=SeawaterIntrusionUnderIce.ipynb. The ISSM software package is publicly available for download from https://issm.jpl.nasa.gov/. The subglacial intrusion melt parameterization will be included in future releases.

## Appendix A: Numerical solutions for seawater intrusion distance on hard beds

In this study, we present various analytic approximations in certain limits for the seawater intrusion distance. The seawater intrusion distance is defined as the location where $h(x) = 1$ governed by the ordinary differential equation

$$\left(Fr^2 - 1\right)\frac{\partial h}{\partial x} = Fr^2\left[\widetilde{C}_i(1-h)^{-1} + \widetilde{C}_d(1+\gamma h)\right] - \Theta \tag{A1}$$

where $Fr = Fr_0 h^{-3/2}$ and we have a single boundary condition $h(x=0) = Fr_0^{2/3}$. This problem is an initial value problem where the unknown is the value of $x$ at which a certain criteria is satisfied. Therefore, it is simple enough to integrate the above equation numerically, starting from $x=0$ and continuing to march towards lowers value of $x$ until the criteria is satisfied.

However, there is one aspect of this problem which requires care in the marching scheme, namely that $Fr(x=0)=1$ introduces a singularity in $dh/dx$. Though there are sophisticated ways to handle such an issue analytically, to facilitate a straightforward numerical solution, we simply set $Fr(x=0)=1-\sqrt{\epsilon}$ where $\epsilon=10^{-16}$ is a machine-precision perturbation to the Froude number, and use a variable resolution marching scheme, where

$$dx = \epsilon + (dx_0 - \epsilon)\tanh\left[\left(\frac{dh}{dx}\right)^{-1}\right], \tag{A2}$$

where $dx_0$ is a larger step size (we use $0.1$). This approach will ensure this method accurately captures the boundary layer near $x=0$ where $h$ changes rapidly and the step size needs to be quite small, while using a more computationally efficient large step size $dx_0$ away from the boundary layer where $h$ is changing more slowly.

*Author contributions.* AAR and EW conceived the study based on EW's prior work. AAR and HS modified ISSM and AAR conducted the ISSM simulations. AAR wrote the manuscript with contributions from all authors.

*Competing interests.* The authors declare that they have no conflict of interest.

*Acknowledgements.* Thanks to Winnie Chu, Adrian Jenkins, Matthew Siegfried, Samer Naif, Joyce Sim, Jeremy Bassis, Peter Washam, Chris Chungkei Lai and Vincent Verjans for helpful discussions during the completion of this work. The final product was improved by helpful reviews from Carolyn Begeman and an anonymous reviewer, in addition to editing from Nicolas Jourdain and Nanna Karlsson. Computing resources were provided by the Partnership for an Advanced Computing Environment (PACE) at the Georgia Institute of Technology, Atlanta. We are thankful for PACE Research Scientist Fang (Cherry) Liu's assistance on HPC challenges. This work was funded in part by startup support from the Georgia Institute of Technology and the University System of Georgia.

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
