# Peer review of "Layered seawater intrusion and melt under grounded ice"

_The Cryosphere, 2021_

## Author Comment (AC1)

**Response to Reviewer #1 for Manuscript "Layered seawater intrusion and melt under grounded ice" by Robel, Wilson, and Seroussi**

Robel et al. present a simplified model of a subglacial salt wedge and apply a simple parameterization of associated subglacial melting in ice sheet models to estimate the effects of this process on sea-level projections. They offer a valuable extension of the work recently presented by Wilson et al. (2020) from intrusion in subglacial channels to intrusion in a subglacial water film. Overall, I found the manuscript clearly written, with some additional suggestions for improving clarity provided below. The primary concern I'd like to see the authors address, elaborated below, pertains to the feasibility of the simulated melt rates.

Our gratitude to this reviewer for their confidence in our concept and their thoughtful comments. We have addressed them each below.

Major comments:

When you discuss the conditions for maintaining stratification between layers, showing that the Reynolds number indicates a laminar regime would be helpful. (Roughly around L240)

Added a parenthetical note indicating that the Reynolds number in such a case would be approx 50, which is well below the laminar-turbulent transition.

A table of some of the cases you discuss in Sections 3.1 and 3.2, i.e., the parameter combinations and distance estimates, would be a useful reference.

We have added such a table (Table 1 in new version)

It seems that you consider the bed slope but not the ice slope in Equations 3,4. However, I think you also make the assumption that ice base slope is the same as bed slope such that the subglacial film thickness is constant. I think that's fine since you treat the velocity of the upper layer as a free parameter, but I'd like to see Equations 3,4 presented in as general a form as possible.

Slope in the ice base that is different in slope of the bed would produce variations in total layer thickness (H) that would be captured through the barotropic pressure gradient already in equations 3 and 4. Ultimately, such changes affect both layers equally and so do not have an impact on the intrusion distance (as we explain with respect to the barotropic pressure gradient). We have added some discussion to this effect specifically referencing the possibility that the ice base and bed slopes are different.

Can you provide an argument that such intrusion-induced melt rates can exist in a steady state? Given such thin films, I would think it's possible that heat exchange (in this laminar regime) from the open ocean upstream through the subglacial layer might be too slow to maintain these melt rates.

This is a nice suggestion. We have done this calculation and added a new paragraph to this effect at the beginning of section 4. Generally, we find that for melt rates of order 10's of m/yr over intrusion

distances of hundreds to thousand of meters (and the range of layer thicknesses explored in section 3), an ocean current of cm/s would be needed to maintain a steady-state in the two-layer model. This is within the range of ocean current speeds measured near the grounding lines of ice streams in West Antarctica, therefore we find this to be plausible. However, this does raise a good point, that such steady-state considerations do place a limit on the melt rates and intrusion distances that can be maintained in a steady-state with seawater intrusion. We have discussed all of these issues in the new paragraph. We have also added a new suite of MISMIP+ simulations with lower baseline basal melt rates in response to a suggestion of reviewer #2 which should also help address this comment.

L540: This line reads like a recommendation. However, with such a wide range of intrusion distances and the strong sensitivity to the bed type, this recommendation seems too general to be useful to ice sheet modelers. Can you settle instead on something like, "uncertainty in intrusion-induced melt should be incorporated into uncertainty of sea-level rise projections using ice sheet models" or "sensitivity of sea-level rise projections to intrusion-induced melt should be tested in ice sheet models." Or something even more specific like "given large uncertainties, we recommend applying melt to partially-grounded cells"

Good point. We have changed this to: "Additionally, the sensitivity of sea level projections to uncertainties in intrusion-induced melt should be tested in a deliberate fashion (rather than as a numerical artifact)."

Minor comments:
L7. Since you haven't yet discussed what you mean by "hard bed" I recommend instead calling it an impermeable bed.
Fixed

L11. "10-50% higher or 100% higher" This is a strange way of expressing it without indicating what makes the difference between the two cases.
Split into two sentences and clarified

L12: "whether the conditions are met for extensive seawater intrusion" or "whether extensive seawater intrusion occurs"
Fixed

L50: "distance" >> "extent"
Fixed

L89: "the bulk porosity of the sheet" Wouldn't it be better to define $\phi_1$ and $\phi_2$ for each layer in Equations 3,4 so that the equations are more general? And similarly for $c_d$ to acknowledge that the ice and bed could have different roughnesses?
We have changed these to different porosities, drag coefficients and obstacle diameters in equations 3 and 4, but then explicitly stated what assumptions we are making about drag coefficients and differences between layers after that. This should make equations 3 and 4 very general for those who might want to start from those later for more complicated scenarios.

L91: This is the final term in Equation 3 but not Equation 4
Fixed

L106: "vertical interface" To me, this is unclear and would be better stated just as the horizontal extent of the saline layer.
Fixed

L107: "without considering their compositional differences" Are you referring to the inclusion of the buoyancy (reduced gravity) term? Or the representation as a two-layer system?
Both, added a parenthetical clarification

L113: "after $H_2$ is eliminated" This would make more sense if $h = H_1/H$ had already been introduced.
Moved the non-dimensionalization up before this, and explained the process of reaching this final equation in more detail.

L116: I think it would be helpful to describe what $\gamma$ represents qualitatively (i.e., porosity).
Description added

L118: "$Fr = Fr_0 h^{-2/3}$" should be explained
Explanation added

Figure 1. I think it would be helpful to include a second panel that is the schematic of the soft bed case.
Second panel added with soft bed case indicated by hatching.

Figure 2. Define h in the caption.
Added definition (also fixed y label)

L191: "single hydraulic potential" implies that Equation 12 will contain the hydraulic potential rather

than $U_{in}$.
Removed this phrase to clarify the statement further.

L195: Can you direct readers to where in the literature $\alpha$ is defined?
Added

L222: "which" is ambiguous between $g'$ and the density difference.
Fixed

L246: I think you mean "it is still possible to maintain two layers in water sheets 10cm thick"?
Yes, fixed

L250: Provide a reference for maximum packing density.
Realized in the process of adding this that we want the maximum density in 3D (which is equivalent to porosity), not 2D. This lower the maximum expected gamma to 2. Added a reference to the "Kepler conjecture" which establishes the maximum packing density of spheres in 3D.

L252: "That could be supported in such a thin water sheet" by virtue of what? Do you mean supporting two layers within the film? How did you determine the $c_d$ value of 0.01? If it is just the upper bound in the literature, then it would be clearer to delete the clause I reference here.
Clause deleted and clarified

L265-272: This paragraph seems to be a distraction. As far as I can tell, the key point is that the layer in the ice sheet interior is on the order of mm and thus too thin for the 2-layer model to apply. That could be stated in the previous paragraph with the Engelhardt and Kamb (1997) reference.
Good point. We have removed the paragraph and added a sentence to the previous paragraph to this effect.

L294: delete "equation"
Fixed

Figure 3: It is hard to view inset box in panels a,c. It might be worth stating in the caption that the blue region is above the critical slope.
Added

Figure 3 and 4: I wonder if you might find a better colormap for these plots. It's pretty hard to see the

difference between values just above and below 10m. Another option would be a contour at 100m since that is the lowest value you implement in the ice sheet model.

Contour added indicating 100 m intrusion distance.

L341: Add units to K

Added

L346: "perhaps in under" >> "perhaps under"

Fixed

Figure 6: Add a legend to panel b for the two points located at x=0

Legend added

L478: The local bed type isn't easy to accomplish in a model if we still don't have a continent-wide map of hard/soft bed.

Modified this sentence to reflect this uncertainty in bed type

L491: "act to prevent strong curvature" This isn't clear offhand. Can you provide a reference?

This was perhaps more speculative. The sentence has been deleted for clarity.

L531: "rights" >> "right"

Fixed

L533: "under the right circumstances
repetitive with previous sentence

Deleted

---

## Author Comment (AC2)

**Response to Reviewer #2 for Manuscript "Layered seawater intrusion and melt under grounded ice" by Robel, Wilson, and Seroussi**

This study tackles a very critical concept in the world of ice sheet-ocean interactions which to date has not received a great deal of attention. Using simplified mathematical descriptions of the near-terminus subglacial environment, it builds on previous work to describe intrusions of warm salty water into the subglacial water layer under marine ice sheets, and establishes that there could potentially be warm water underlying subglacial environments kilometers from the grounding line. It then considers, through a simple parameterisation, how the presence of this water, if able to effect high melt rates compared that that observed in ice-shelf cavities, might impact marine ice sheet stability. In most respects the study is comprehensive, well thought out and well explained, and feel it deserves publication. I have two main comments for the authors and editor to consider, followed by a number of minor ones.
Thank you to this reviewer for their thorough and helpful suggestions, which are each addressed below.

Main Comments:
1) While the hard-bed treatment of the subglacial layer is very detailed and well explained, and one can clearly see how physical considerations lead to mathematical results, I feel the soft-bed case just presents results without intuition. I appreciate that soft-bed transport is probably not the mode by which subglacial transport occurs, given the volumes involved – but it would be nice to understand, if only qualitatively, where 12-14 come from without needing to read the Strack paper referenced.
We have added an explanation of the key assumptions that go into these equations and the general approach to the derivation (without a complete derivation, which is already done in other papers that we reference). In particular we discuss the Dupuit-Forcheimer and Ghyben-Herzberg approximations which form the basis for much of saltwater intrusion theory in hydrology, how they are combined at the outset, and eventually leading to the equations for intrusion distance through flat or sloped aquifers.

2) The parameterisation for under-ice melt in (17) and illustrated in Fig 5 (and was actually also used by Parizek et al 2013 – and in fact was required by that study to show instability of Thwaites) is acknowledged to be just a parameterisation with little oceanographic basis – but I still feel it is a bit of a cop-out to say that we don't have the understanding to rigorously model melt so we will just use an approach that is linear in space, as an exploratory tool. This method still leads to extremely high melt rates under grounded ice, which is known (from sources cited in the manuscript) to have a much larger impact of grounded ice than melt of similar magnitudes under ice shelves. I am certainly not an expert in boundary-layer oceanography but I know that the melt rates suggested by e.g. the equations of Holland and Jenkins 1999 under ice shelves exposed to CDW require quite developed boundary layers and high levels of turbulent mixing, and Im unsure if such mixing rates and layer thicknesses are allowed by the theory. The examples cited (Kimura and Begeman) invoke double diffusion – but I believe that in both of these works, the under-ice ocean conditions below the ice would lead to much higher melt rates than observed considering under-shelf plume type flow. I bring this up not because I understand

the physics of flow in these subglacial environments – but simply because the analogues cited to justify these high melt rates actually show very very low melt rates under such stratification envisioned in this study, and it is difficult to imagine what could make them higher. It would be quite a lot to ask the authors to come up with a better parameterisation, or to redo their experiments. But i feel, and if the editor agrees, that the use of such a parameterisation should be far more heavily caveated than it is, and in more places in the text than just its introduction.

The reviewer followed up with the following comment:
I am unable to amend my comment, so i hope that the authors and editor read this as well!

I realised that my General Comment #2 could be interpreted as saying, if under-ice melt rates were of similar magnitude to those seen in Kimura et al and Begeman et al they would be unimportant. This is my intention in any way. Under-ice sheet melting of a few m/a near the grounding line would be on the order of thinning rates in some of the fastest-thinning ice streams in Antarctica. However, this is a bit overshadowed by considering melt rates of 30-40 meters per year, which for reasons given in my original comment are difficult to imagine on physical grounds.

We respond to both the reviewer's original comment and the addendum (inserted above for clarity) here.

The point raised by the reviewer is an excellent one. Ultimately, we do not address the physics of heat fluxes in this manuscript since it would require a more sophisticated treatment of the ice-water boundary layer than what we are able to sketch out with a relatively idealized mathematical model. In order to get at the basal melt rates with some accuracy, ultimately a more sophisticated fluid dynamical model which simulates turbulent mixing and heat fluxes would be required. Such simulations are actually currently being undertaken by a student, but are well outside the scope of this manuscript. We have added additional caveats in section 4 and in the discussion to this end. We have also added a reference to Parizek, which does indeed explore the sensitivity of simulations to melt just upstream of the grounding line.

We do seriously take the point that the melt rates of 10's of m/yr considered in these ISSM simulations are higher than would be expected for double-diffusive convection. The reason why these melt rates were originally used is because they are in line with the standard MISMIP+ benchmarks which we are trying to reproduce here to provide a point of comparison for others who are attempting to discern the importance of intrusion-induced melt in a "standard" configuration. Additionally, such melt rates are also in line with those observed (and typically modeled) at Thwaites. So, we have retained these original simulations, but also added new ones to address this point. Given that we argue for the potential for intrusion-induced melt through double-diffusive convection, even in the absence of strong interfacial mixing, it is worth it to determine whether the results hold up for melt rates more typical of such a stratified sub-ice layer. Thus, we have added a suite of new MISMIP+ simulations (Figure 7) where

the "baseline" case has basal melt rates at the grounding line an order of magnitude lower (i.e. a few m/yr) in line with melt rates observed where there is double-diffusive convection (e.g., Kimura et al. 2015). We find in these simulations that though the absolute ice volume loss is lower (as expected for lower melt rates in the floating ice), intrusion-induced basal melt over hundreds to thousands of meters upstream of the grounding line still increases volume loss by 10-55%. While these increases are lower than in the higher baseline melt scenarios, it is still the case that intrusion-induced melt can have first-order effect on the rate of ice volume loss.

We also point out (in response to the reviewer's comment) that basal melt rates of 10's of m/yr are within the range of melt rates that have been observed near the fastest-melting grounding lines, and that since ice flow generally compensates for most basal (and surface) melting, the corresponding net thinning rates would be much lower. Indeed, in our transient simulations with high basal melt rates, the thinning rates are generally in the range of 0.1-1 m/yr. Thus, even though the highest thinning rates in Antarctica are m/yr, basal melt rates may still be considerably higher than this.

Minor comments:
Line 92: this is not the final term in eq 4
Fixed

line 92 Rominger reference: it would still be nice to see intuition for the functional form of this term
Added further explanation of this new term

Derivation of (5), and expressoin for Fr: I think some clarity is needed. do you not require eqs 1 and 2 as well (or at least eq 1)? How is $H2$ cancelled, is it via $H2 = H - H1$? The nondimensionalisatoin seems to imply H is a constant, rather than a varying field.. if this is the case, i can't find where it is made clear.
This discussion has been reorganized a bit (also in response to suggestions by reviewer 1), which should clarify how this expression is reached (by combining equations 1-4 after non-dimensionalization). Have also added explanation that H is indeed a constant (though $H_1$ and $H_2$ are not).

line 138 hard$-->$difficult
Fixed

line 146: what is W
Removed W and added words to describe instead

Figure 2: labels (a) through (d) not shown. what are the parameters in eq 5 for these solutions other

Labels added and other parameters indicated in caption.

paragraph at line 218: awkward. shouldn't the solutions to eq 5 be *exactly* this length scale, as you are implicitly defining the length scale through this equation?
Not necessarily - as the length scale is derived through approximation of the exact intrusion distance that is the solution to equation 5. It is easy to see how this paragraph is confusing on this point, however, and we have re-written it to clarify the point that we are making (just that there are only a few parameters determining intrusion distance).

line 246: *stratification within* water sheets?
Deleted "stratified" to clarify

line 274: what does "exponentiated" mean in this context? does L become an exponent somewhere?
Deleted to clarify and simplify this sentence

line 287 2nd "of till" redundant
Fixed

References
Holland, D. M., & Jenkins, A. (1999). Modeling thermodynamic iceâocean interactions at the base of an ice shelf. Journal of Physical Oceanography, 29(8), 1787-1800.

Parizek, B. R., et al. (2013), Dynamic (in)stability of Thwaites Glacier, West Antarctica, J. Geophys. Res. Earth Surf., 118, 638â 655, doi:10.1002/jgrf.20044.